# Bandwidth-control orbital-selective delocalization of *4f* electrons in epitaxial Ce films

Yi Wu[1], Yuan Fang[1], Peng Li[1], Zhiguang Xiao[1], Hao Zheng[1], Huiqiu Yuan ⓘ [1,2,3], Chao Cao ⓘ [4], Yi-feng Yang ⓘ [5,6,7✉] & Yang Liu ⓘ [1,2,3✉]

The 4f-electron delocalization plays a key role in the low-temperature properties of rare-earth metals and intermetallics, and it is normally realized by the Kondo coupling between 4f and conduction electrons. Due to the large Coulomb repulsion of 4f electrons, the bandwidth-control Mott-type delocalization, commonly observed in d-electron systems, is difficult in 4f-electron systems and remains elusive in spectroscopic experiments. Here we demonstrate that the bandwidth-control orbital-selective delocalization of 4f electrons can be realized in epitaxial Ce films by thermal annealing, which results in a metastable surface phase with reduced layer spacing. The quasiparticle bands exhibit large dispersion with exclusive 4f character near $\bar{\Gamma}$ and extend reasonably far below the Fermi energy, which can be explained from the Mott physics. The experimental quasiparticle dispersion agrees well with density-functional theory calculation and also exhibits unusual temperature dependence, which could arise from the delicate interplay between the bandwidth-control Mott physics and the coexisting Kondo hybridization. Our work opens up the opportunity to study the interaction between two well-known localization-delocalization mechanisms in correlation physics, i.e., Kondo vs Mott, which can be important for a fundamental understanding of 4f-electron systems.

[1] Center for Correlated Matter and Department of Physics, Zhejiang University, Hangzhou, China. [2] Zhejiang Province Key Laboratory of Quantum Technology and Device, Zhejiang University, Hangzhou, China. [3] Collaborative Innovation Center of Advanced Microstructures, Nanjing University, Nanjing, China. [4] Department of Physics, Hangzhou Normal University, Hangzhou, China. [5] Beijing National Laboratory for Condensed Matter Physics, Institute of Physics, Chinese Academy of Sciences, Beijing, China. [6] School of Physical Sciences, University of Chinese Academy of Sciences, Beijing, China. [7] Songshan Lake Materials Laboratory, Dongguan, Guangdong, China. ✉email: yifeng@iphy.ac.cn; yangliuphys@zju.edu.cn

Rare earth metals and intermetallics are prototypical strongly correlated electron systems and host a variety of interesting quantum phases, including heavy fermion, quantum criticality, unconventional superconductivity, etc[1–5]. Key to understand these exotic phases lies at unraveling the peculiar nature of the 4f-electron delocalization, which is often described by the periodic Anderson model (PAM) composed of localized 4f electrons and dispersive conduction bands. The 4f electrons can be delocalized by the many-body Kondo effect, giving rise to an effective hybridization with conduction electrons and a flat quasiparticle band near the Fermi energy ($E_F$). Such a hybridized band picture is often employed to describe realistic 4f-based materials[6–8]. By contrast, in transition metal systems, d electrons typically have appreciable dispersions due to large intersite hopping and their localization/delocalization follows the well-known Mott mechanism as described in the single or multi-band Hubbard model[9]. It is therefore intriguing to think if in 4f-based materials, the Mott physics from 4f electrons may exist and cooperate with Kondo hybridization to cause correlated phenomena beyond the standard theoretical framework.

Unfortunately, since the 4f bandwidth is often negligible compared to the onsite Coulomb repulsion U, direct experimental evidence for the Mott-type delocalization is very rare (if any) for 4f electrons. The most notable 4f-electron system where the bandwidth-control orbital-selective delocalization (OSD) might occur is the pure Ce metal, where the Ce atoms are ordered in closed-packed structures with smallest possible f–f distance[10]. In fact, the 4f-electron Mott delocalization was proposed early on to explain the famous first-order γ–α transition of Ce[11–14], but an alternative Kondo volume collapse (KVC) scenario was also proposed, attributing the γ–α transition to a large increase in the hybridization strength between conduction and 4f electrons (c–f hybridization)[15–18]. Some theoretical studies further pointed to the importance of the interplay between two mechanisms[19–23]. While previous photoemission studies provided compelling evidences that the Kondo effect plays an important role in the electronic structure of Ce[24–30], no spectroscopic evidence supporting the Mott scenario has ever been reported. Unambiguous identification of the bandwidth-control 4f delocalization is experimentally challenging, as the coexisting Kondo screening could lead to somewhat similar itinerant 4f bands. Nevertheless, there are clear differences in their 4f quasiparticles: in the bandwidth-control Mott scenario, 4f quasiparticle bands can exhibit large dispersion with nearly pure 4f character, while those in the Kondo picture can acquire dispersion only through hybridization with nearby conduction bands, leading to mixed orbital characters. Here in this paper, we present direct spectroscopic evidence for the bandwidth-control OSD of 4f electrons in epitaxial Ce films, by combining high-quality thin film growth by molecular beam epitaxy (MBE) and in situ measurements from angle-resolved photoemission spectroscopy (ARPES). In particular, we uncovered a hitherto unreported metastable phase of Ce, where sharp dispersive quasiparticle bands with pure 4f character can be observed near $E_F$, accompanied by an appreciable dispersion of the lower Hubbard band. Such spectral character, which cannot be explained by the Kondo hybridization picture, can be well accounted for by the bandwidth-control Mott physics.

## Results

**Tuning across the bandwidth-control OSD of 4f electrons.** The single-crystal Ce films are grown by MBE on epitaxial graphene layers and in situ ARPES measurements were performed immediately after film growth (see "Methods"). Figure 1a summarizes the ARPES spectra of one thick Ce film taken at ~20 K; these

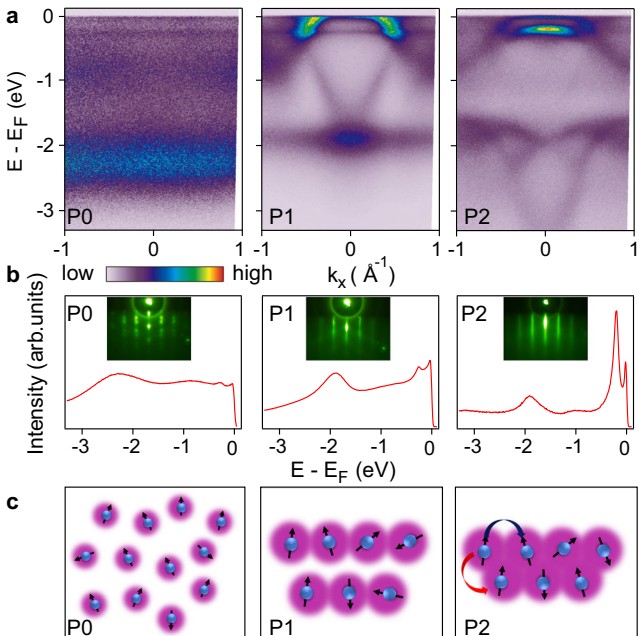

**Fig. 1 Electronic structure of a thick Ce film at 20 K after progressive post-growth annealings from ~370 to ~540 K. a** Three representative ARPES spectra at different stages of annealings: disordered Ce film (P0), ordered Ce films at intermediate annealing temperature (P1), and at high annealing temperature (P2). **b** Energy distribution curves (EDCs) at the $\bar{\Gamma}$ point for these three phases. The insets are the corresponding RHEED images. **c** Cartoons illustrating the proposed origin of the different electronic structures. The reduction of the interlayer spacing in P2 is exaggerated.

three spectra were taken after sequential annealings (from ~370 to ~540 K) after film deposition (see Supplementary Fig. 1 for more data). Three distinct electronic phases, labeled P0, P1, and P2, can be readily identified. P0 is a polycrystal Ce film, confirmed from reflection high energy electron diffraction (RHEED) measurements shown in Fig. 1b. It features three non-dispersive bands that are well described by single-impurity Anderson model (SIAM)[24,26]: the lower Hubbard band (or $4f^0$) at −2.2 eV, weak $4f^1_{5/2}$ and $4f^1_{7/2}$ Kondo resonance peaks at $E_F$ and −0.25 eV, respectively. The broad peak at −0.9 eV is likely due to the unbinding 5d electrons. The Kondo resonance peaks near $E_F$ do not show any momentum dependence (expected for a polycrystal sample), implying that these resonance states remain localized at each individual Ce site.

Upon annealing to intermediate temperature, an ordered single-crystal Ce film can be obtained (P1), as determined by streaky diffraction pattern in RHEED and clear dispersive conduction bands in the ARPES spectra. This phase was observed in previous ARPES studies and identified as the γ-phase[29–31]. Compared to the P0 phase, the lower Hubbard band shifts to slightly lower energy (−1.9 eV) and remains dispersionless, and the Kondo resonance peaks near $E_F$ become sharper and more pronounced. The quasiparticle bands near $\bar{\Gamma}$ (Fig. 2) feature a reversed-U-shaped band with a large effective mass, characteristic of the c–f hybridization expected from the hybridized band picture within PAM (see below)[32,33].

Further annealing to higher temperature leads to the P2 phase, which has not been reported before. It maintains a highly ordered structure with similar in-plane lattice constant as P1. Now the lower Hubbard band at ~−1.8 eV develops clear dispersion (Fig. 1b), indicating that the intersite hopping of the originally localized 4f electrons is suddenly turned on (Fig. 1c). The

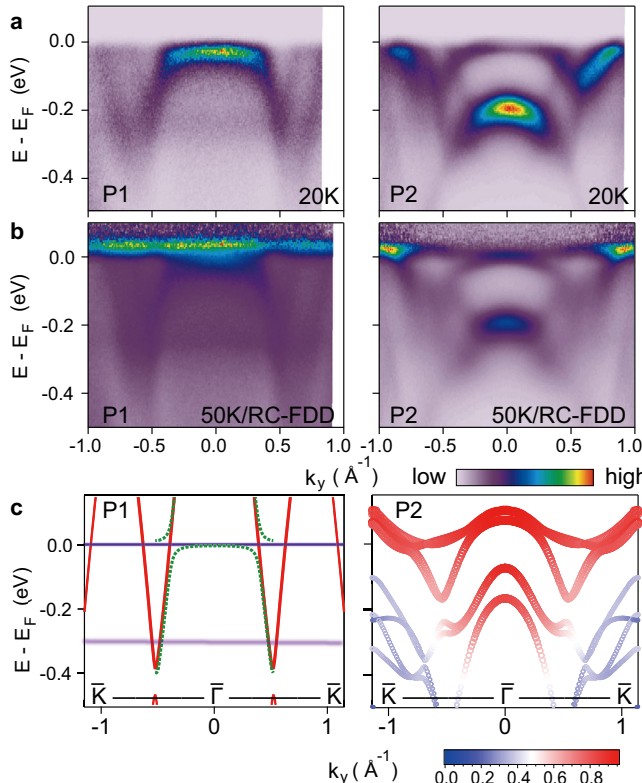

**Fig. 2 Quasiparticle dispersion of the P1 and P2 phases near $E_F$. a** ARPES spectra taken at 20 K along $\bar{\Gamma}\bar{K}$ for P1 (left) and P2 (right). **b** ARPES spectra divided by RC-FDD, taken at 50 K along $\bar{\Gamma}\bar{K}$ for P1 (left) and P2 (right). **c** Band structure calculations along $\bar{\Gamma}\bar{K}$ for comparison with the experimental data of P1 (left) and P2 (right). In the left panel, red curves are conduction band dispersion from localized $4f$ calculation from DFT, and horizontal cyan lines at $E=0$ and $E=-0.25$ eV indicate the Kondo resonance peaks from SIAM: they hybridize to form the reversed-U-shape band observed experimentally. Dashed green curves are simulated band dispersion based on the hybridized band approach within PAM. The right panel shows the result from itinerant $4f$ calculation, where the color of the plotted bands indicates the $4f$ orbital contribution (scale bar at the bottom).

resulting increase of the lower Hubbard bandwidth has a profound effect on the quasiparticle bands near $E_F$, which can no longer be described by the hybridized band picture within PAM as the P1 phase (see below). This is a direct manifestation of the bandwidth-control delocalization of $4f$ electrons expected from the Mott–Hubbard model. Since the $4f$ spectral function still consists of two parts similar to P1, i.e., a lower Hubbard band that remains far from $E_F$ and the quasiparticle bands that lie very close to $E_F$, the P2 phase is not in a strongly mixed-valent state.

**Quasiparticle dispersions and Fermi surfaces across the P1 → P2 transition**. A detailed comparison of the quasiparticle dispersions of the P1 and P2 phases is shown in Fig. 2a (see Supplementary Fig. 2 for more data). The quasiparticle bands of the P1 phase can be well described by the hybridized band picture within PAM: away from $E_F$, the overall dispersion of conduction bands is in reasonable agreement with the "localized $4f$" calculation from density-functional theory (DFT) (Fig. 2c), which simply treats $4f$ electrons as core electrons; near $E_F$, however, the Kondo resonance peaks emerge and the $c$–$f$ hybridization leads to the characteristic reversed-U-shaped band expected for a hole-band crossing. A simple simulation (with a hybridization strength $V = 70$ meV adapted from a recent study[29]) can reproduce the

experimental results well (green dashed curves in Fig. 2c). Transition to the P2 phase leads to dramatic changes in the quasiparticle dispersion: first, the originally flat $4f_{7/2}^1$ peak is replaced by an intense hole-like band at $\sim-0.2$ eV near $\bar{\Gamma}$, with clear dispersion. Since the localized $4f$ calculation does not yield any conduction band near this energy–momentum region (Supplementary Fig. 3), this hole-like band should mostly come from the $4f$ orbitals. Such dispersive quasiparticle band with pure $4f$ character cannot be explained by the same hybridization picture as the P1 phase. Instead, it is a signature of the delocalized $4f$ quasiparticles expected from the Mott–Hubbard model. Another important difference between the two phases is that the P2 phase develops a sharp W-shaped band right near $E_F$, which again cannot be accounted for by the simple $c$–$f$ hybridization.

To reveal the full spectral function near $E_F$, we divide the ARPES spectra by the resolution-convoluted Fermi-Dirac distribution (RC-FDD), which are shown in Fig. 2b. The results highlight the distinct behavior of the $4f$ bands for these two phases: the $4f$ band in P1 is very flat and shows only slight bending upon crossing the conduction bands, as expected for a Kondo resonance; by contrast, the $4f$ bands in P2 show obviously larger dispersion with very fine structures. Surprisingly, the observed $4f$ bands for the P2 phase can be very well described by "itinerant $4f$" calculations from DFT (Fig. 2c), where the $4f$ electrons are taken into account similarly as other conduction electrons. The good agreement between experiment and calculation indicates that the $4f$ electrons in P2 develop into coherent band-like quasiparticles near $E_F$. Orbital analysis further confirms that the dispersive quasiparticle bands near $E_F$ (close to $\bar{\Gamma}$) are derived almost entirely from $4f$ orbitals, while those away from $E_F$ with larger in-plane momentum begin to mix with the conduction bands, reflecting the coexistence of Kondo hybridization.

The transition from the Kondo regime (P1) to the $4f$ Mott-delocalized regime (P2) leads to dramatic change in the Fermi surfaces (FSs), as shown in Fig. 3. The FS of the P1 phase features a hole pocket with $4f$ character centered at the $\bar{\Gamma}$ point, with strong intensity expanding over a large radius $\sim0.5$ Å$^{-1}$ due to the flatness of heavy quasiparticle bands. The FS also has elliptical pockets centered at the $\bar{M}$ point, which originate from shallow $4f$ bands at the $\bar{M}$ point (Supplementary Fig. 2). By contrast, the FS of the P2 phase exhibits a small pocket at the $\bar{\Gamma}$ point and large triangular pockets as a result of the W-shaped band along $\bar{\Gamma}\bar{K}$ (Fig. 2a). The distinct FS shapes highlight the different means how the $4f$ electrons participate the FS: one forms heavy bands near $E_F$ through the $c$–$f$ hybridization, while the other develops coherent band-like quasiparticles well explained by DFT. Away from $E_F$, the constant energy contours for the P1 phase are dominated by conduction bands, showing a hole pocket centered at $\bar{\Gamma}$ and electron pockets centered at $\bar{M}$. The off-$E_F$ contours for the P2 phase are similar to P1, except that the pockets at $\bar{M}$ are more circular and the sharp $4f$ band at $\sim-0.2$ eV gives rise to an additional strong pocket at zone center. This again demonstrates that the quasiparticle bands in the Mott mechanism span a larger energy window compared to those from the Kondo physics, which is typically constrained near $E_F$.

**Origin of the P1 → P2 transition**. Now we address the physical origin of the P1–P2 transition. We first emphasize that this transition is caused by different annealing temperatures of Ce films, as the ARPES measurements in Figs. 1 and 2 were performed at identical temperatures (20 or 50 K). It would be natural to think that the isostructural $\gamma$–$\alpha$ transition (face-centered cubic, fcc) could take place with lowering temperature[34], which can be dependent on the annealing conditions and therefore account for

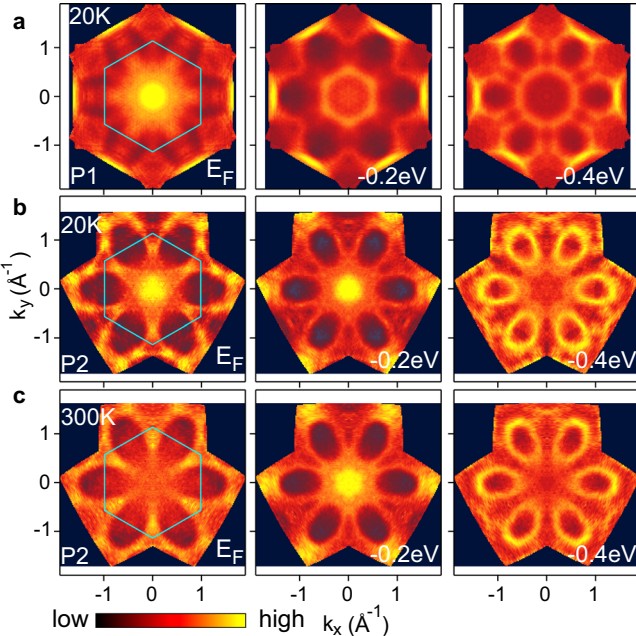

**Fig. 3 Constant energy contours for the P1 and P2 phases. a, b** Constant energy contours at $E = 0$ (i.e., the FS), −0.2 and −0.4 eV for P1 (**a**) and P2 (**b**) at 20 K. **c** Constant energy contours for P2 at 300 K. The Brillouin zone is indicated by the cyan hexagon.

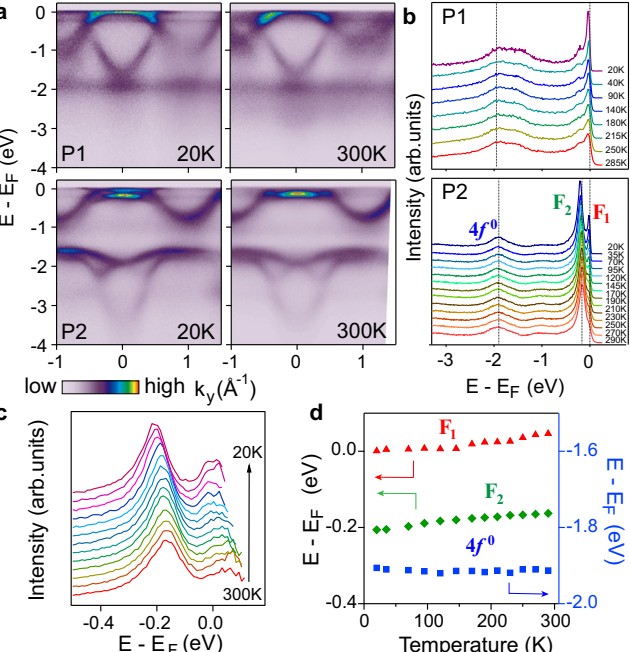

**Fig. 4 Temperature evolution of the electronic structure. a** ARPES spectra of P1 (top) and P2 (bottom) along $\bar{\Gamma}\bar{M}$ at 20 and 300 K. **b** The temperature evolution of EDCs at $\bar{\Gamma}$. The $4f$ peaks for P2 are labeled individually. **c** Temperature-dependent EDCs divided by RC-FDD for P2. **d** Temperature evolution of the peak positions for P2.

the observed P1–P2 transition. However, our in-plane FS maps in Fig. 3a, b imply that the in-plane lattice constant is very close for these two phases (within the experimental uncertainty). Our temperature-dependent ARPES measurements also show no sign of any sudden electronic transition expected for the γ–α transition with ~5% change in the lattice constant (see Fig. 4a, b and Supplementary Fig. 3). Figure 5 summarizes the results from ex situ temperature-dependent X-ray diffraction (XRD) measurements, which rule out any temperature-driven structural transition for both phases in the ARPES temperature range (20–300 K). For P1, we found that only ~1% of the film converts to the α-phase starting at ~10 K (Fig. 5a), which is very different from the expected γ–α transition at ~150 K for bulk Ce[34]. We mention that the similar suppression of the γ–α transition in epitaxial Ce films has also been reported[29,35], although the percentage of the α-phase at low temperature differs due to different growth conditions.

A clear structural difference between P1 and P2 is that for well annealed P2 samples, additional diffraction peaks at ~31° and ~65° emerge and persist in the entire temperature range (Fig. 5b). This indicates that an unreported structural phase exists in P2, with a layer spacing ~3.5% smaller than the bulk γ. Note that the peak intensity from this phase is small (~2%) compared to that of the dominant γ-phase, and it increases with further annealing (Supplementary Fig. 4). Since its lattice constant does not correspond to any known bulk phase of Ce[10,36], it is most likely a metastable phase that forms at the surface and grows with progressive annealing. Indeed, it is well-known that the surface structure can be different from the bulk, due to altered atomic bonds and associated energy minimization. Such metastable surface structure normally requires certain threshold of thermal energy (i.e., temperature) to overcome the local kinematic barrier, possibly resulting in reduced interlayer spacing[37,38]. To check if a metastable surface structure with reduced layer spacing could indeed be possible in Ce films, we performed structural relaxation calculations of a freestanding γ-phase slab (Fig. 6a). The calculation showed that the outmost Ce bilayer has a reduced

layer spacing by ~0.13 Å, i.e., ~4.4% reduction compared to the bulk γ. The metastable surface structure could also involve a change of layer stacking sequence: in particular, the hexagonal closed-packed (hcp) β-phase is the other possible low-temperature phase of Ce, which differs from the γ-phase only in the layer stacking sequence (ABC for γ and ABAC for β)[39,40]. Similar calculation considering the ABAC stacking suggested a similarly reduced layer spacing by ~4.7%, with even lower surface energy compared to relaxed γ layers (Fig. 6a).

Taken together, our results indicate that a metastable surface structure with reduced layer spacing (~3.5%) forms near the surface upon annealing, leading to the observed P1–P2 transition. A ~3.5% reduction in the layer spacing, considered to be large for strongly correlated $4f$-electron systems, could induce the intersite $f$–$f$ hoping, resulting in the transition from the standard Kondo lattice (P1) to the bandwidth-control delocalized state (P2), as illustrated in Fig. 1c. Indeed, the itinerant $4f$ calculation, which uses the experimental structural parameters, yields good agreement with the experimental data for the P2 phase (Figs. 2c and 6b, c). In particular, the W-shaped band near $E_F$, the strong hole band at ~−0.2 eV, and the conduction band at −3.8 eV are well reproduced.

**Temperature evolution of the electronic structure.** For the P1 phase, we observe no sudden temperature-driven transition in the electronic structure (Fig. 4a, b), and only the intensity of the Kondo peaks near $E_F$ gradually diminishes with increasing temperature (Supplementary Fig. 5), expected for a Kondo system. Note that the Kondo peaks persist up to high temperature, similar to other Ce-based Kondo systems[33,41], which is likely due to its high coherence temperature (~90 K from ex situ transport results, see Supplementary Fig. 6) and Kondo screening possibly involving excited crystal electric field (CEF) states[33,41–43]. While inelastic neutron scattering suggests an excited CEF quartet at ~17 meV above the ground state doublet[44], we do not observe any

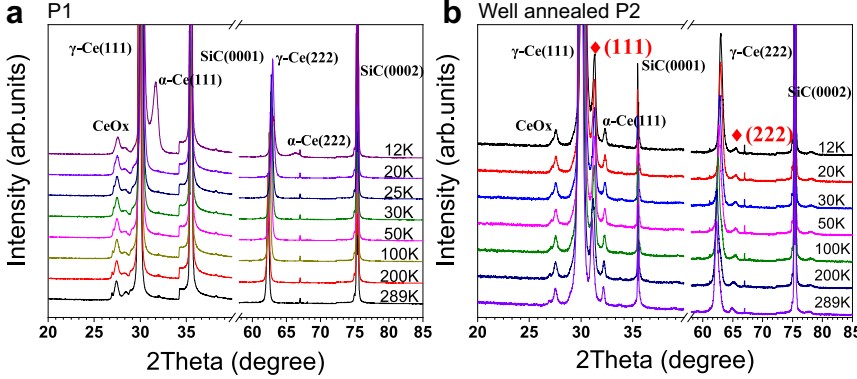

**Fig. 5 Temperature-dependent XRD measurements.** Ex situ temperature-dependent XRD results for P1 (**a**) and P2 (**b**). All peaks are labeled individually. The red stars in **b** highlight the diffraction peaks at ~31° and ~65°, corresponding to the metastable surface phase. The peak at ~28° corresponds to Ce oxides due to (inevitable) partial oxidation. The tiny peak at ~33° in **b** implies possibly a very small mixture of the α-phase (see Supplementary Fig. 4).

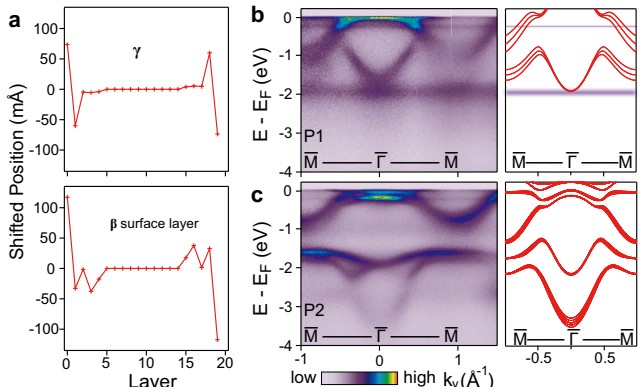

**Fig. 6 The electronic transition from P1 to P2. a** Out-of-plane structural relaxation of the surface γ or β layers in a 20 atomic-layer Ce slab, whose inner layers are fixed to be the bulk γ-phase. The shifted positions refer to $z$ positions relative to the expected bulk value. **b, c** Experimental valence band dispersion (left) of P1 (**b**) and P2 (**c**) along $\bar{\Gamma}\bar{M}$, in comparison with DFT band structure calculations (right). In the P1 calculation, diffuse cyan lines at $E = -1.9$, $-0.25$, and $0$ eV indicate the lower Hubbard band, the $4f^1_{7/2}$ and $4f^1_{5/2}$ Kondo peaks, similar to Fig. 2.

clear satellite in the Kondo resonance peaks, likely due to the large width and limited energy resolution. For the P2 phase, we observe gradual temperature-dependent evolution of the quasiparticle peaks (Fig. 4b–d), which takes place over a wide temperature window. Specifically, the quasiparticle peak at $\sim -0.2$ eV (F2) moves by $-0.04$ eV from 300 to 20 K, and a similar shift occurs for the F1 peak near $E_F$, which becomes clear after the energy distribution curves are divided by the corresponding RC-FDDs (Fig. 4c, d). On the other hand, the lower Hubbard band at $\sim -1.8$ eV does not show noticeable shift with temperature.

The temperature-dependent band shift in the P2 phase, absent in P1, could be a manifestation of the interplay between the Mott and Kondo physics. Detailed analysis shows that the conduction bands exhibit a slight downward shift with decreasing temperatures (Supplementary Fig. 7). Following the Luttinger theorem, this implies a small but discernible charge transfer between the 4f and conduction bands[45]. For typical Ce-based Kondo lattice systems in the Kondo regime, such as Ce-115 systems (and the P1 phase), the Ce valence is always close to 3+ and shows negligible temperature dependence[46]. It is hence attempted to associate the observed shift with the more itinerant character of 4f electrons and their hybridization with conduction bands. We emphasize that the change is nonetheless tiny and the lower Hubbard band

in the P2 phase remains sufficiently far from $E_F$, implying that the system is unlikely in a strongly mixed-valent state.

Despite the temperature-dependent band shift, the overall quasiparticle bands at high temperature remain similar to the low temperature case. Specifically, the F2 quasiparticle peak remains strong and well-defined up to 300 K (the F1 peak shows similar behavior after division by RC-FDD, see Fig. 4c), exhibiting very little intensity decrease with increasing temperature. This demonstrates that quasiparticles at elevated temperatures maintain the key characters that arise from the Mott physics[47].

## Discussion

Our results show that the bandwidth-control delocalization from the Mott mechanism is indeed possible for 4f electrons, at least in the densely packaged Ce. It is interesting to note that the experimental bandwidth of the lower Hubbard band appears to be ~0.5 eV in the P2 phase (Fig. 6c), although this could be underestimated due to overlapping conduction bands. How such a seemingly small bandwidth could trigger the electronic transition requires further study in the future. We mention that some residual Kondo resonance peaks can be observed in P2 (Figs. 4a and 6b, c), indicating that the Kondo effect arising from the Ce³⁺ local moments is still in play here. Therefore, it would be natural to expect that the coexisting Kondo effect could play a role in the P1–P2 delocalization process, although the delicate interplay between the Kondo effect and delocalization in the Mott channel remains an open question. It is interesting to note that a dispersive lower Hubbard band was reported before for a α-like Ce monolayer on W(110), although the dispersion of quasiparticle bands near $E_F$ was not available[48].

The OSD of 4f electrons observed here bears some analogy to the orbital-selective Mott transition (OSMT) proposed in multi-orbital ruthenates[49–51]. Nevertheless, there are obvious differences: (1) There is much stronger tendency towards localization for 4f electrons compared to 4d electrons (due to larger $U$ and smaller bandwidth). (2) The bands involved in OSMT in ruthenates, i.e., $d_{xy}$ and $d_{xz}/d_{yz}$ orbitals, do not hybridize due to different wavefunction symmetries, but the 4f and conduction electrons in Ce exhibit clear hybridization, i.e., Kondo hybridization, which very likely cooperates with the bandwidth-control Mott physics.

The bandwidth-control OSD in Ce is most likely caused by reduced layer spacing near the surface, supported by our ex situ XRD measurements and DFT calculations, although more in situ study, e.g., from surface X-ray scattering, is still needed to understand the detailed structure. Theoretically, the reduced layer spacing at the surface reduces the crystal symmetry (from fcc to

hcp), which in turn changes the CEF splitting and broadening. The altered CEF states could further affect the Kondo screening and possibly the bandwidth-control delocalization. More studies are needed in the future to resolve the fine structures in the quasiparticle bands associated with the CEF states. Finally, the possibility of stabilizing such surface phase in appreciable volume (detectable by bulk-sensitive XRD and transport measurements) offers exciting opportunities in the future to study the physical properties associated with the 4f Mott physics.

To summarize, by combining MBE growth and in situ ARPES measurements, we were able to track the evolution of the electronic structure of epitaxial Ce films under different growth conditions and measurement temperatures. The electronic structure with sequential annealings undergoes transition from the P0 phase (described by SIAM) to the P1 phase (described by the hybridized band approach within PAM), and finally to the P2 phase (described by the Mott physics with coexisting Kondo process). Our work provides direct evidence for the bandwidth-control OSD of 4f electrons, which was studied extensively by theory but lacks spectroscopic proof. The bandwidth-control delocalization and Kondo effect conspires to give rise to an intriguing electronic phase with coherent 4f quasiparticle bands, which agree well with itinerant 4f calculation and exhibit unusual temperature dependence. Our results demonstrate that the Kondo and Mott mechanisms can coexist (and even act cooperatively), in some Kondo systems with small f–f distance, which could be important for a fundamental understanding of 4f-electron systems. Our observation in Ce thin films may also help clarify the mechanism of the γ–α transition in bulk Ce.

## Methods

**MBE growth and in situ ARPES measurements.** Thick Ce films (typically >10 nm) were grown by MBE in a growth chamber with a pressure $<5 \times 10^{-11}$ mbar. The deposition was controlled with a high-temperature effusion cell to yield a rate of approximately 2 Å/min, as determined by a quartz crystal monitor mounted near the sample. Highly doped 6H-SiC(0001) substrates were used to obtain epitaxial graphene layers by heating up to ~1400 °C in ultrahigh vacuum. The Ce films were grown on top of these epitaxial graphene layers at a temperature of ~100 °C. After the deposition, the films were annealed sequentially (from ~370 up to ~540 K) to achieve different electronic phases as shown in Fig. 1. RHEED patterns were monitored during the process to check the film quality in real time. ARPES measurements were performed immediately after the film growth, by transferring the sample under ultrahigh vacuum from the MBE chamber to the connected ARPES chamber. All ARPES spectra were taken with a sample temperature of ~20 K, unless noted otherwise. The base pressure of the ARPES system was $7 \times 10^{-11}$ mbar, which increased to ~$2.3 \times 10^{-10}$ mbar during the Helium lamp operation (beam size ~1 × 1 mm²). All the ARPES data were taken with He-II photons (40.8 eV, corresponding to $k_z \sim \pi$ for an estimated inner potential ~10 eV), where the photoemission cross section for the 4f electrons is appreciable. Due to the low counts from He-II photons, we used an analyzer pass energy of 10 eV, which yielded an overall energy (momentum) resolution of ~15 meV (~0.01 Å⁻¹). The RC-FDD, used for recovering the spectral function near $E_F$ in Figs. 2b and 4c, was obtained by fitting an Au reference scan taken under identical measurement condition.

**DFT calculations.** Electronic structure calculations were performed using DFT based on the projected augmented wave method including the spin–orbit coupling, as implemented in the Vienna ab initio simulation package (VASP). For the itinerant 4f calculations, the Ce-4f electrons are included as the valence states; while for the localized 4f calculations, the Ce-4f are treated as core electrons, similar to the open-core treatment in full-potential linearized augmented plane-wave calculations. These two calculations are often carried out in Ce-based systems to represent two limiting situations of Ce-4f electrons (fully itinerant vs localized), see for example ref.[52]. An energy cutoff of 437 eV (300 eV) for itinerant (localized) 4f calculations was employed, with the **k** mesh being 17 × 17 × 17(16 × 16 × 4) for the fcc (hcp) phase. The localized 4f calculation for P1 adopted the bulk-like γ structure. For the P2 phase, we found that itinerant 4f calculations within DFT yield very good agreement with the ARPES data. This is perhaps not too surprising as dynamic mean-field theory calculations show reasonably good agreement with DFT calculations in the itinerant 4f-electron regime, after considering a renormalization factor[53,54]. The itinerant 4f calculations for P2 used the hcp structure with reduced interlayer spacing determined from ex situ XRD (Fig. 5b) to simulate the surface electronic structure probed experimentally. Calculations for both AB-stacking (its existence on bulk form remains debated[10]) and ABAC-stacking

(the naturally stable β-phase[39,40]) hcp phases are performed and they yield overall similar results (Supplementary Fig. 8). The result from the ABAC-stacking β-phase is adopted in the manuscript (Figs. 2c and 6c), as this configuration has a lower total energy.

**Ex situ XRD and transport measurements.** Ex situ XRD and transport measurements were performed for very thick films (typically ~200 nm), after the films were characterized by in situ ARPES measurements. To minimize sample oxidation, the XRD and transport measurements were done immediately after they were taken out of the vacuum chamber.

## Data availability

All the data supporting the findings are available from the corresponding authors upon reasonable request.

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

## Acknowledgements

We thank Yabin Liu, Prof. Guanghan Cao, Dr. Xiaohe Miao, and Prof. Hangdong Wang for assistance with the low-temperature XRD measurements, and Ding Wang, Hang Su, Lichang Yin, and Chufan Chen for ex situ transport measurements. This work is supported by the National Key R&D Program of the MOST of China (Grant No. 2016YFA0300203, 2017YFA0303100), the National Science Foundation of China (No. 11674280, 11974397, 11974306, and 12034017), the Key R&D Program of Zhejiang Province, China (2021C01002), the Science Challenge Project of China (No. TZ2016004), and the Strategic Priority Research Program of the Chinese Academy of Sciences (Grant No. XDB33010100).

## Author contributions

The project was designed by Y.L. Thin film growth and in situ ARPES measurement was performed by Y.W., with help from P.L. and H.Z. ARPES data analysis was done by Y.W., Y.-f.Y. and Y.L. DFT calculations were carried out by Y.F. and C.C. Ex situ transport and XRD measurements were performed by Y.W., with help from Z.-G.X. and H.-Q.Y. The manuscript was prepared by Y.W., Y.-f.Y., and Y.L. All authors discussed the results and commented on the manuscript.

## Competing interests

The authors declare no competing interests.

## Additional information

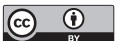 

