## [Peer Review File · Nature Communications]

REVIEWER COMMENTS

Reviewer #1 (Remarks to the Author):

The paper of Wu et al. reported an interesting experimental discovery of a novel phase of Ce materials in epitaxial thin Ce film. The ARPES data shows an intriguing interplay of the Kondo lattice physics, as described by the periodic Anderson model, and the Mott physics, as described by the Hubbard model. The work provides the first direction experimental evidence of bandwidth-controlled orbital-selective Mott transition in f-electron systems. The transition from a gamma-like phase (referred to as P1 in the manuscript) to this novel phase, upon annealing, was explained by the shortened inter-layer distances and consequently the enhanced inter-site hopping amplitude between the Ce atoms. The observed phenomena and the possible driving mechanism provide a new avenue for further experimental and theoretical investigations that have significant implications for 4f-electron systems.

The paper is well written and properly structured. The phenomenon reported in the paper and the intriguing underlying physics is of great interest to the electronic structure community. I thus strongly recommend the publication of the manuscript in Nature Communication.

That said, I nevertheless have some minor comments on the DFT calculations presented in the paper. In a number of locations, the authors mentioned that they performed “localized 4f calculations” or “itinerant 4f calculations” of DFT. I am wondering what really characterized these two types of calculations? Are these constrained DFT calculations, or they are done with different initializations? Namely, how the “localized” or “itinerant” characters of 4f electrons are enforced within the calculations? Some elaborations of this point would be helpful.

Reviewer #2 (Remarks to the Author):

The manuscript of Yi Wu et al., reports on a ARPES study of epitaxial Ce films. Dependent on annealing methodology two different phases are identified. The first phase, labeled P1, is already discussed in previous studies (Ref. 29,30). A new phase, labeled P2, is identified and reported here for the first time. The films hosting this new P2 phase appears to be of better quality judge from the RHEED and photoemission data. Moreover, the band structure probed by the ARPES experiment is different in P1 and P2. This band structure difference is interpreted in terms of enhanced hopping

probability (in P2) due to shorter lattice parameters. This interpretation has a number of shortcomings as will be pointed out in following comments. This combined with the overlap of existing photoemission work on Ce films suggest that a more specialized (or slightly lower impact) journal would be more appropriate for this work. At least, a sharpening of this manuscript is necessary before publication in nature communications can be considered.

Comments and suggestions:

(a) The interpretation of the data obtained on the two phases P1 and P2, draws analogues to Mott insulating physics often found transition metal oxides. However, both P1 and P2 are metallic phases with Fermi surfaces as shown in Figure 3. A Mott insulating state should not have a Fermi surface. It is therefore misleading to claim a band width controlled Mott transition based on the presented data.

(b) From the data, it might make sense to speak about an orbitally selective delocalization that could be tuned by the lattice parameter (and thereby the band width). This scenario is, however, not directly connected to a Mott insulating state. In best case, it would be linked to certain orbital selective Mott insulating states proposed for the ruthenates. In all cases, if the authors wish to use analogues to Mott insulating states, it would be best to connect to a multi-orbital scenario.

(c) In first place, the band-width control interpretation seems to emerge from comparison between data and DFT calculations. This comparison (Fig. 5) is not overly impressive. In fact, there is no or little reason to believe that DFT band structure calculations would capture a strongly correlated electron system. Therefore, it would be important with additional experimental evidence of a 3% lattice parameter change. Would ex-situ x-ray diffraction be a possibility.

Reviewer #3 (Remarks to the Author):

The manuscript by Wu et al studies epitaxial Ce films using in-situ ARPES and DFT calculations. They find Kondo hybridized f states that transition into a Mott delocalized state as the sample is annealed to high temperatures, related to a reduction of out of plane lattice constant. The nature of f-electrons in heavy fermion compounds, whether localized or itinerant, plays a key role in their electronic properties. The study here demonstrates, in some sense, a transition from partially

localized f electronic states to more delocalized states. The provided data is of high quality. However, I have a few questions that need to be addressed.

The main question here is how to distinguish between a Kondo Hybridized states and Mott delocalization. In figure 2, the authors represent their ARPES data. In both cases, a flat band is observed at the Fermi energy, with the possible distinction that in the P2 phase, the f level (Kondo resonance from the f5/2) is shifted to slightly above EF. Usually, the position of the f-level relative to EF is related to the valence of the f-electrons. This raises the question whether the P1 to P2 transition is related to any valence change? Perhaps this can be seen in the temperature dependent data in Fig.4 if the Fermi Dirac function is taken into account. This brings me to my second question.

The temperature dependence of the f5/2 Kondo resonance in the P1 phase does not seem to follow the expected T-dependence $\Gamma \propto \sqrt{(\pi k_B T)^2 + (k_B T_K)^2}$. Rather it seems almost constant. The authors claim this is similar to other Ce-based heavy fermions such as Ce115. However, it is known from ARPES and STM data that the both the amplitude and width of the Kondo resonance change rapidly below the coherence temperature of about 50-100K (for example: Phys. Rev. B 96, 045107). Do the authors have resistivity measurements showing the coherence temperature in this system?

On the other hand, the temperature evolution of the F1 peak in the P2 phase does show a rapid T-dependence below 100K. If this is a true reduction in the amplitude, it may represent a Kondo resonance. On the other hand, if this is related to temperature dependent shift of the f-level to above the Fermi energy, then dividing with the Fermi Dirac function may show the shift at high temperature. Can the authors show which case it is?

The authors show that the surface layer can shrink due to the annealing, while the in-plane lattice constant remains unchanged. This by itself breaks the crystal symmetry changing the crystal field effects. The authors should address this in the manuscript.

In Fig.1 P1 phase shows a V like conduction band above -2eV. This band seems to be gone in the P2 phase. The DFT calculations in S2 do not show the disappearance of this band. Can the authors comment on this?

Finally, the emergence of the f5/2 and f7/2 states in both P1 and P2 phases are a consequence of the Kondo effect, meaning that in both phases there should be some level of localized moments. This may indicate that the system in the P2 phase is not fully itinerant yet.

Minor issues:

The authors clearly demonstrate that the observed transition is a surface effect and not a property of the bulk. This does not change the importance of the work, however, the authors need to clearly state that in their abstract/title.

The color scale in Fig.1a P0 phase is saturated. Can this be changed? Also, the authors claim that

“these states remain localized at each individual Ce site and hence show no momentum Dependence”. However, this is a polycrystalline sample. So what is the size of the ARPES beam and is it expected from a polycrystals to see momentum dependence?

Re: NCOMMS-20-38164

Authors: Yi Wu, *et al.*

Dear reviewers,

Thank you very much for your useful comments. Following your suggestions, we have revised our manuscript substantially. This includes:

1. Replacing Fig. 1(a), Fig. 2(c), Fig. 4(b-d) and Fig. 6(b) with new ones; adding new figures (Fig. 2(b), Fig. 5, Fig. S2, Fig. S4, Fig. S5, Fig. S6 in supplementary information (SI)). Figure captions are changed or added accordingly. The relevant supplementary figures are now mentioned explicitly in the manuscript text for easy reading.
2. Changing the title, abstract, and relevant texts in the manuscript.

Below we address the questions in a point-by-point fashion (**All the revised parts in the manuscript and SI are colored in red**).

Reviewer 1:

The paper of Wu et al. reported an interesting experimental discovery of a novel phase of Ce materials in epitaxial thin Ce film. The ARPES data shows an intriguing interplay of the Kondo lattice physics, as described by the periodic Anderson model, and the Mott physics, as described by the Hubbard model. The work provides the first direction experimental evidence of bandwidth-controlled orbital-selective Mott transition in f-electron systems. The transition from a gamma-like phase (referred to as P1 in the manuscript) to this novel phase, upon annealing, was explained by the shortened inter-layer distances and consequently the enhanced inter-site hopping amplitude between the Ce atoms. The observed phenomena and the possible driving mechanism provide a new avenue for further experimental and theoretical investigations that have significant implications for 4f-electron systems. The paper is well written and properly structured. The phenomenon reported in the paper and the intriguing underlying physics is of great interest to the electronic structure community. I thus strongly recommend the publication of the manuscript in Nature Communication.

Our response:

We thank the reviewer for his/her positive comment on our paper.

That said, I nevertheless have come minor comments on the DFT calculations presented in the paper. In a number of locations, the authors mentioned that they performed “localized 4f calculations” or “itinerant 4f calculations” of DFT. I am wondering what really characterized these two types of calculations? Are these constrained DFT calculations, or they are done with different initializations? Namely, how the “localized” or “itinerant” characters of 4f electrons are enforced within the calculations? Some elaborations of this point would be helpful.

Our response:

We thank the reviewer for this suggestion. To clarify this issue, we have added the following explanations to the revised manuscript:

1. (page 3, 4th paragraph): “away from E_F , the overall dispersion of conduction bands is in reasonable agreement with the 'localized 4f' calculation from density-functional theory (DFT) (Fig. 2(c)), which simply treats 4f electrons as core electrons; near E_F , however, the Kondo resonance peaks emerge and the c - f hybridization leads to the characteristic reversed-U-shaped band expected for a hole-band crossing.”;
2. (end of page 3): “itinerant 4f” calculations from DFT (Fig. 2(c)), where the 4f electrons are taken into account similarly as other conduction electrons. The good agreement between experiment and calculation indicates that the 4f electrons develop into coherent band-like quasiparticles near E_F .”;
3. (Method, DFT calculations, 3rd line): “The localized and itinerant 4f calculations were performed by choosing different pseudopotentials of Ce, which treat the Ce 4f electrons as core electrons (localized 4f) or band electrons (itinerant 4f), respectively.”.

Reviewer 2:

The manuscript of Yi Wu et al., reports on a ARPES study of epitaxial Ce films. Dependent on annealing methodology two different phases are identified. The first phase, labeled P1, is already discussed in previous studies (Ref. 29,30). A new phase, labeled P2, is identified and reported here for the first time. The films hosting this new P2 phase appears to be of better quality judge from the RHEED and photoemission data. Moreover, the band structure probed by the ARPES experiment is different in P1 and P2. This band structure difference is interpreted in terms of enhanced hopping probability (in P2) due to shorter lattice parameters. This interpretation has a number of short comings as will be pointed out in following comments. This combined with the overlap of existing photoemission work on Ce films suggest that a more specialized (or slightly lower impact) journal would be more appropriate for this work. A least, a sharpening of this manuscript is necessary before publication in nature communications can be considered.

Our response:

We thank the reviewer for the comment. While we agree with the reviewer that there have been previous ARPES studies on the P1 phase discussing the Kondo hybridization in Ce films, we do not think that these existing papers undermine the novelty and importance of our current paper. The central and new discovery of our paper is the direct observation of Mott-type bandwidth-control orbital-selective delocalization in a 4f-electron system by ARPES, which has not been reported before to the best of our knowledge. This is realized by the discovery of the new P2 phase. We emphasize that the experimental electronic structure of Ce-based Kondo lattice systems is generally interpreted from the Kondo hybridization picture (as in P1), and the bandwidth-control Mott mechanism (which plays a key role in d-electron systems) is often overlooked. Although the possible involvement of Mott physics in 4f-electron systems was discussed extensively by theory (Ref. [11-14, 19-23]), direct spectroscopic signature remains elusive. Therefore, our observation of the Mott physics in Ce provides an exciting platform to understand the interplay between two important delocalization mechanisms in correlated electron systems (Mott vs Kondo), which can be important for a fundamental understanding of 4f-electron systems.

Following the reviewer's suggestion, we do think that it is necessary to sharpen the writeup of the original manuscript to highlight the importance of our work. Therefore, we have revised our manuscript accordingly, which includes changes in the following parts:

1. (abstract): "The 4f-electron delocalization plays a key role in the low-temperature properties of rare-earth metals and intermetallics, including heavy fermions and mix-valent compounds, and is normally realized by the many-body Kondo coupling between 4f and conduction electrons. Due to the large onsite Coulomb repulsion of 4f electrons, the bandwidth-control delocalization from the Mott physics, commonly observed in d-electron systems, is difficult in 4f-electron systems and remains elusive in spectroscopic experiments."
2. (page 2, beginning of 2nd paragraph): "Unfortunately, since the 4f bandwidth is often negligible compared to the onsite Coulomb repulsion U , direct experimental evidence for the Mott-type delocalization is very rare (if any) for 4f electrons."
3. (page 2, end of 2nd paragraph): "In particular, we uncovered a hitherto unreported metastable phase of Ce, where sharp dispersive quasiparticle bands with pure 4f character can be observed near EF, accompanied by an appreciable dispersion of the lower Hubbard band. Such spectral character, which cannot be explained by the Kondo hybridization picture, can be well accounted for by the bandwidth-control Mott physics."
4. (Last paragraph in "discussion" section): "Our work provides direct evidence for the bandwidth-control OSD of 4f electrons, which was studied extensively by theory but lacks spectroscopic proof."

Comments and suggestions:

(a) The interpretation of the data obtained on the two phases P1 and P2, draws analogues to Mott insulating physics often found transition metal oxides. However, both P1 and P2 are metallic phases with Fermi surfaces as shown in Figure 3. A Mott insulating state should not have a Fermi surface. It is therefore misleading to claim a band width controlled Mott transition based on the presented data.

Our response:

We thank the reviewer for this point. Indeed, the Mott-insulating state of the P1 phase refers to the localized character of the 4f orbitals (if the Kondo hybridization is not taken into account), while in P2 the 4f orbitals become delocalized by themselves due to direct f-f hopping, similar to correlated d-electron systems. As the reviewer correctly pointed out, it should be better called an orbital-selective delocalization of 4f electrons.

To clarify this issue (the bandwidth-control orbital-selective delocalization) and avoid possible confusion for the readers, we have made the following revisions:

1. We have changed the title, which now reads "Bandwidth-control orbital-selective delocalization of 4f electrons in epitaxial Ce films"
2. We have made corresponding changes to the abstract and relevant texts to emphasize that the P1-P2 transition is a bandwidth-control orbital-selective delocalization of 4f electrons. Specifically, this includes:
 - a. (abstract, middle): "bandwidth-control orbital-selective delocalization of 4f electrons"
 - b. (Page 1, 1st paragraph, second line from bottom): "the Mott physics from 4f electrons may"

- c. (page 1, 2nd paragraph): “where the **orbital-selective delocalization (OSD)** might occur”, “4f-electron Mott **delocalization**”, “Unambiguous identification of the **bandwidth-control 4f delocalization** is experimentally challenging”, “direct spectroscopic evidence for the bandwidth-control **OSD** of 4f electrons”
- d. (page 2, “Results” subtitle): “Tuning across the **bandwidth-control OSD of 4f electrons**”
- e. (page 4, 2nd paragraph): “from the Kondo regime (P1) to the **4f Mott-delocalized regime (P2)**”
- f. (“discussion” section, 3rd paragraph): “The bandwidth-control **OSD** in Ce is most likely...”

(b) From the data, it might make sense to speak about an orbitally selective delocalization that could be tuned by the lattice parameter (and thereby the band width). This scenario is, however, not directly connected to a Mott insulating state. In best case, it would linked to certain orbital selective Mott insulating states proposed for the ruthenates. In all cases, if the authors wish to use analogues to Mott insulating states, it would a best connect to a multi-orbital scenario.

Our response:

We thank the reviewer for this suggestion. As discussed in the reply to question (a), the P1-P2 transition refers to an orbital-selective delocalization of 4f electrons, although the system remains metallic due to contribution from non-4f conduction electrons. In some sense, the current case is analogous to the orbital-selective Mott transition proposed in ruthenates. Following the reviewer’s suggestion, we have substantiated the discussion on this point, to emphasize the multi-orbital scenario and the connection to other multi-orbital systems:

1. We have made changes to **the abstract, the introduction and discussion sections**, to emphasize that the P1-P2 transition is an orbital-selective transition (see reply to question (a)).
2. (“discussion” section, 2nd paragraph): “**The OSD of 4f electrons observed here bears some analogy to the orbital-selective Mott transition (OSMT) proposed in multi-orbital ruthenates [50, 51, 52]. Nevertheless, there are obvious differences: 1. There is much stronger tendency towards localization for 4f electrons compared to 4d electrons (due to larger U and smaller bandwidth); 2. The bands involved in OSMT in ruthenates, i.e., d_{xy} and d_{xz}/d_{yz} orbitals, do not hybridize due to different wavefunction symmetries, but the 4f and conduction electrons in Ce exhibit clear hybridization, i.e., Kondo hybridization, which very likely cooperates with the bandwidth-control Mott physics.**”

(c) In first place, the band-width control interpretation seems emerge from comparison between data and DFT calculations. This comparison (Fig. 5) is not overly impressive. In fact, there is no or little reason to believe that DFT band structure calculations would capture a strongly correlated electron system. Therefore, it would be important with additional experimental evidence of a 3% lattice parameter change. Would ex-situ x-ray diffraction be a possibility.

Our response:

We thank the reviewer for the suggestion regarding additional *ex-situ* X-ray diffraction (XRD) measurements. In the past few months, we have performed extensive temperature-dependent XRD measurements and finally found experimental evidence of a ~3.5% layer-spacing reduction from P1 to P2. The results are summarized in new Fig. 5 and Fig. S4 in SI. The trick here is to anneal the P2 phase thoroughly (note that the P2 phase has a wide temperature window where it

shows similar ARPES spectrum), so that the metastable surface phase with reduced layer spacing can develop from the surface and grow with progressive annealing, ultimately allowing for detection by XRD. We mention that for mildly annealed P2 phase (ARPES spectrum shows similar P2 feature), peaks associated with the contracted surface phase do not show up in XRD (Fig. S4), probably due to a small thickness if not annealed thoroughly. This provides experimental support of our previous proposal: the P2 phase starts from the surface region and expands in thickness/volume upon progressive annealings. We mention that the XRD peaks corresponding to the contracted surface phase remain small compared to the dominant γ -phase peaks ($\sim 2\%$ for well annealed samples), implying that this surface phase takes up only a small portion of the entire film. Nevertheless, the possibility of stabilizing the P2 phase in appreciable volume (at least detectable by bulk-sensitive XRD) opens up exciting opportunities to study its physical properties in details (see, e.g., the new transport results shown in Fig. S6).

We have made the following changes to the manuscript, to include our new XRD results and discuss their implications:

1. We have included a new figure (Fig. 5) in the manuscript, as well as a new figure in the SI (Fig. S4), for the new temperature-dependent XRD results. The figure captions are changed accordingly.
2. We have revised the section “Origin of the P1 \rightarrow P2 transition” substantially to discuss the XRD results. This includes:
 - (a) (1st paragraph): “Figure 5 summarizes the results from ex-situ temperature-dependent X-ray diffraction (XRD) measurements, which rules out any temperature-driven structural transition for both phases in the ARPES temperature range (20-300 K). For P1, we found that only $\sim 1\%$ of the film converts to the α phase starting at 10 K (Fig. 5(a)), which is very different from the expected γ - α transition at ~ 150 K for bulk Ce [35].”
 - (b) (2nd paragraph): “A clear structural difference between P1 and P2 is that for well annealed P2 samples, additional diffraction peaks at $\sim 31^\circ$ and $\sim 65^\circ$ emerge and persist in the entire temperature range (Fig. 5(b)). This indicates that a new structural phase exists in P2, with a layer spacing $\sim 3.5\%$ smaller than the bulk γ . Note that the peak intensity from this new phase is small ($\sim 2\%$) compared to that of the dominant γ phase, and it increases with further annealing (supplementary Fig. S4). Since its lattice constant does not correspond to any known bulk phase of Ce [10,37], it is most likely a metastable phase that forms at the surface and grows with progressive annealing.”
 - (c) (2nd paragraph): “To check if a metastable surface structure with reduced layer spacing could indeed be possible in Ce films”.
 - (d) (3rd paragraph): “Taken together, our results indicate that a metastable surface structure with reduced layer spacing ($\sim 3.5\%$) forms near the surface upon annealing, leading to the observed P1-P2 transition.”
 - (e) (3rd paragraph): “Indeed, the itinerant 4f calculation, which uses the experimental structural parameters, yields very good agreement with the experimental data for the P2 phase (Fig. 2(c) and Fig. 6(b)).”
3. We have replaced the DFT calculations for P2 (Fig. 2(c) and Fig. 6(b)) with the new results using the experimentally determined lattice constant. The new results are close to the previous results, since the experimental lattice constant is actually close to the value obtained from DFT simulation.
4. The details of ex-situ XRD and transport results are now included in a new subsection in

“Methods”, called “Ex-situ XRD and transport measurements”.

Reviewer 3:

The manuscript by Wu et al studies epitaxial Ce films using in-situ ARPES and DFT calculations. They find Kondo hybridized f states that transition into a Mott delocalized state as the sample is annealed to high temperatures, related to a reduction of out of plane lattice constant. The nature of f-electrons in heavy fermion compounds, whether localized or itinerant, plays a key role in their electronic properties. The study here demonstrates, in some sense, a transition from partially localized f electronic states to more delocalized states. The provided data is of high quality. However, I have a few questions that need to be addressed.

Our response:

We thank the reviewer for his/her positive comment.

The main question here is how to distinguish between a Kondo Hybridized states and Mott delocalization. In figure 2, the authors represent their ARPES data. In both cases, a flat band is observed at the Fermi energy, with the possible distinction that in the P2 phase, the f level (Kondo resonance from the $f_{5/2}$) is shifted to slightly above E_F . Usually, the position of the f-level relative to E_F is related to the valence of the f-electrons. This raises the question whether the P1 to P2 transition is related to any valence change? Perhaps this can be seen in the temperature dependent data in Fig.4 if the Fermi Dirac function is taken into account.

Our response:

We thank the reviewer for this suggestion. We followed the reviewer’s suggestion and added more data/analysis to reveal the 4f bands better. This includes:

1. A new figure (Fig. 2(b)) showing the ARPES spectra divided by resolution-convoluted Fermi-Dirac distribution (RC-FDD), which recover the full spectral function near E_F . The old Fig. 2(a) is now moved to SI as Fig. S2. The new data in Fig. 2(b) clearly shows the difference of the 4f dispersion between P1 and P2: the 4f band in P1 is very flat and only shows slight bending upon crossing the conduction band, expected for a Kondo resonance; by contrast, the 4f bands in P2 show obviously larger dispersion (even in regions where no conduction bands can be found), in good agreement with the itinerant 4f calculation in Fig. 2(c). This provides direct experimental evidence for a transition from the typical Kondo lattice (P1) to the bandwidth-control delocalized state (P2).
2. The temperature-dependent EDCs divided by RC-FDD for the P2 phase are now shown in Fig. 4(c). The peak positions are summarized in Fig. 4(d). The results show that the F1 peak indeed show very small shift with temperature, similar to the F2 peak at -0.2 eV. This implies that the valence in P2 can change slightly with temperature – A crude estimation based on the Luttinger volume (and assuming a spherical k-space shape) yields a valence change of ~ 0.03 from 300 K to 20 K. This is likely a direct consequence of the interplay between bandwidth-control delocalization and the coexisting Kondo hybridization (see below for more discussion). By contrast, the quasiparticle peaks in P1 do not show any noticeable shift with temperature (see

Fig. S5), implying smaller valence change, as expected for a typical Kondo system close to the Kondo limit.

We agree with the reviewer that it is not straightforward to distinguish the delocalization by Kondo hybridization or bandwidth-control Mott mechanism, particularly if the system is complex or the experimental quasiparticle bands cannot be well resolved. Fortunately, here we are dealing with a simple system with quasiparticle bands that can be well observed experimentally. Specifically, there are three key observations that allow us to make the distinction between the Kondo hybridized case (P1) and the bandwidth-control delocalized case (P2):

1. The experimental quasiparticle dispersion and orbital character is different for the two cases (see Fig. 2(a,b)): In the Kondo hybridized phase (P1), the Kondo resonance peak near E_F acquires dispersion only through hybridization with nearby conduction bands, yielding heavy quasiparticle bands with large effective mass (typically found in heavy fermion systems); in the Mott delocalized phase (P2), the 4f bands near E_F exhibit larger dispersion without the requirement of conduction band crossing, leading to dispersive quasiparticle bands with nearly pure 4f character. This is best illustrated in the RC-FDD data shown in Fig. 2(b).
2. Theoretically, the heavy quasiparticles in the Kondo hybridized phase (P1) can NOT be captured by simple DFT calculations due to the strong local correlation. Often the hybridized band picture within the periodic Anderson model is used for data interpretation (see Fig. 2 and associated discussion). However, the 4f quasiparticles in the bandwidth-control delocalized phase (P2) can be explained by itinerant-4f DFT calculations, subject to possible band renormalization. This is indeed supported by our detailed comparison between experiment and calculation shown in Fig. 2.
3. The evolution of the lower Hubbard band ($4f^0$) provides further evidence for the bandwidth-control delocalization from P1 to P2 (see Fig. 1). Our capability of measuring the in-situ electronic structure at different stage of annealing allows us to track closely the evolution of the electronic structure. Specifically, the lower Hubbard band is very flat (small bandwidth) in P1, but exhibits clear dispersion (large bandwidth) in P2. This indicates that the P1-P2 transition is triggered by the change in the 4f bandwidth, consistent with bandwidth-control delocalization from the Mott mechanism.

As the reviewer suggested, the P1-P2 transition could simultaneously involve some valence change, due to the difference in their 4f band position and temperature evolution. However, the valence change is probably small, since the 4f spectral functions for P1 and P2 are overall similar: both consist of a lower Hubbard band, which remains sufficiently far from the Fermi level (~ -2 eV below E_F), and 4f quasiparticles, which are very close to E_F . Considering a typical U value of ~ 6 eV for 4f electrons, the upper Hubbard band should be well above E_F . This indicates that neither P1 or P2 is in a strongly valence-mixing state – a strongly mixed-valent state would have either lower or upper Hubbard band crossing E_F (or at least in close proximity to E_F). As we discussed in the manuscript, the slightly different valence in P2 is likely due to the cooperative interplay between the Kondo and Mott mechanisms. In other words, the coexisting Kondo screening (a virtual valence-fluctuating process) could cooperate with the bandwidth-control delocalization mechanism and give rise to a small valence change.

To further clarify the difference between P1 and P2, we have made more revisions to the main text, which includes:

1. (abstract): “The resulting quasiparticle bands exhibit large dispersion with exclusive 4f character near $\bar{\Gamma}$ and extend reasonably far below the Fermi energy”.
2. (page 2, 2nd paragraph): “in the bandwidth-control Mott scenario, 4f quasiparticle bands can exhibit large dispersion with nearly pure 4f character, while those in the Kondo picture can acquire dispersion only through hybridization with nearby conduction bands, leading to mixed orbital characters.”
3. (page 3, end of 3rd paragraph): “Since the 4f spectral function still consists of two parts similar to P1, i.e., a lower Hubbard band that remains far from E_F and quasiparticle bands that lie very close to E_F , the P2 phase is not in a strongly mixed-valent state.”
4. (page 3, last paragraph): “To reveal the full spectral function near E_F , we divide the ARPES spectra by the resolution-convoluted Fermi-Dirac distribution (RC-FDD), which are shown in Fig. 2(b). The results highlight the distinct behavior of the 4f bands for these two phases: the 4f band in P1 is very flat and shows only slight bending upon crossing the conduction bands, as expected for a Kondo resonance; by contrast, the 4f bands in P2 show obviously larger dispersion with very fine structures.”

This brings me to my second question. The temperature dependence of the f5/2 Kondo resonance in the P1 phase does not seem to follow the expected T-dependence $\Gamma = \sqrt{(\pi k_B T)^2 + 2 (k_B T_K)^2}$. Rather it seems almost constant. The authors claim this is similar to other Ce-based heavy fermions such as Ce115. However, it is known from ARPES and STM data that the both the amplitude and width of the Kondo resonance change rapidly below the coherence temperature of about 50-100K (for example: Phys. Rev. B 96, 045107). Do the authors have resistivity measurements showing the coherence temperature in this system?

Our response:

We thank the reviewer for the comment. We have followed the reviewer’s suggestion and added new data and analysis to the paper, which includes:

1. A new Fig. 4(b) replacing the old one. The P1 data in the old Fig. 4(b) was taken from one sample and it was used to show that there is no sudden electronic transition in a large temperature window (20-300 K). Unfortunately, due to the unavoidable sample decay during the long temperature-dependent scans (as a result of low counts from He II photons, short in-vacuum life time of chemically reactive Ce and large temperature range needed to cover), the Kondo peak intensity in that figure suffered from such complication and did not show the expected temperature dependence. To overcome this experimental difficulty, we have performed temperature-dependent ARPES measurements from different samples, which were grown under identical condition and covered different temperature regions. This approach allows us to minimize the extrinsic effect from sample decay during the long temperature-dependent scan. The new results, summarized in Fig. 4(b) and Fig. S5, are now consistent with the expectation for a typical Kondo system. The results also agree well with previous reports (ref.[29,30]).
2. The detailed temperature-dependent result of P1 as the new Fig. S5 in the SI (see figure caption for more discussion). This figure shows results from a few representative P1 samples (grown under identical conditions) and summarizes the temperature evolution of the Kondo peak intensity, which follows the $-\log(T)$ behavior expected for a Kondo resonance. This confirms that the observed quasiparticle in P1 is due to Kondo effect. Here we do not present detailed

analysis of the peak width. This is because the experimental width of the Kondo peak is largely dependent on the sample quality (and also truncated by the resolution-convoluted Fermi-Dirac function), which makes it difficult to reliably extract the intrinsic peak width. This is perhaps why most ARPES studies focus on the analysis of the peak position and intensity, instead of the peak width often considered in STM experiments.

3. The new transport data as the **new Fig. S6** in the SI (see **figure caption** for more discussion). The resistivity of the P1 sample clearly shows a broad hump near 90 K, which implies a transport coherence temperature of ~ 90 K.

From the new data and analysis mentioned above, it is clear that the 4f bands observed in P1 are consistent with the typical behavior of Kondo resonance peaks. On the other hand, we do notice that the Kondo peak for the P1 phase can be observed up to room temperature, implying that the Kondo effect takes place already at a very high temperature. Recent studies (e.g., ref. [34]) have shown that the Kondo resonance peaks can be observed at much higher temperature than the transport coherence temperature, due to Kondo screening involving excited crystal electric field (CEF) states. In the current case, the transport coherence temperature (Fig. S6) is ~ 90 K. In addition, the hybridization between the conduction and 4f electrons is strong in P1, supported by a large hybridization strength $V \sim 70$ meV estimated from matching the experimental quasiparticle dispersion (see Fig. 2(c) and also ref. [29]). The strong Kondo effect, which possibly involves excited CEF states, could explain why the Kondo resonance peaks can be observed at such a high temperature.

We have revised the main text in the manuscript to elaborate the understanding of the P1 phase, which now reads: “which is likely due to **its high coherence temperature (~ 90 K from ex-situ transport results, see supplementary Fig. S6) and Kondo screening possibly involving excited crystal electric field (CEF) states [34, 42, 43, 44].**” (page 5, 3rd paragraph).

On the other hand, the temperature evolution of the F1 peak in the P2 phase does show a rapid T-dependence below 100K. If this is a true reduction in the amplitude, it may represent a Kondo resonance. On the other hand, if this is related to temperature dependent shift of the f-level to above the Fermi energy, then dividing with the Fermi Dirac function may show the shift at high temperature. Can the authors show which case it is?

Our response:

We thank the reviewer for this suggestion. In the **new Fig. 4(c)**, we added the temperature-dependent EDCs of the P2 phase, divided by the RC-FDD. As we can see from the data, the F1 peak gradually shifts to higher energy with increasing temperature (**new Fig. 4(d)**), but its intensity does not change much with temperature. This behavior is similar to the F2 peak at -0.2 eV. Such a temperature-dependent shift of quasiparticle bands is different from P1. As discussed in the reply above, the temperature-dependent shift is likely due to the cooperative interplay between the bandwidth-control delocalization and the coexisting Kondo screening. The robust 4f quasiparticles in P2 are consistent with the delocalized quasiparticles from the Mott scenario. To further illustrate this point, we have also made changes to the main text:

1. (page 5, 3rd paragraph): “the quasiparticle peak at ~ -0.2 eV (F2) moves by -0.04 eV from 300 K to 20 K, and **a similar shift occurs for the F1 peak near E_F , which becomes clear after the**

EDCs are divided by the corresponding RC-FDDs (Fig. 4(c-d)).”

2. (page 5, last paragraph): “the F2 quasiparticle peak remains strong and well-defined up to 300 K (the F1 peak shows similar behavior after division by RC-FDD, see Fig. 4(c))”.
3. (page 7, last line in “MBE growth and in-situ ARPES measurements”): “The RC-FDD, used for recovering the spectral function near E_F in Fig. 2(b) and Fig. 4(c), was obtained by fitting a Au reference scan taken under identical measurement condition.”

The authors show that the surface layer can shrink due to the annealing, while the in-plane lattice constant remains unchanged. This by itself breaks the crystal symmetry changing the crystal field effects. The authors should address this in the manuscript.

Our response:

We thank the reviewer for this suggestion. Indeed, the reduced layer spacing at the surface can break the crystal symmetry, from the original fcc lattice to a hexagonal lattice. This means that the Ce $J=5/2$ crystal electric field (CEF) states could change from a doublet + a quartet (in a fcc lattice) to three doublets (in a hexagonal lattice). The altered CEF splitting could in turn affect the Kondo screening and possibly the bandwidth-control delocalization. We note that CEF excitations can in principle lead to satellite peaks in the Kondo resonances (see, e.g., Ref. [34,42,43]). Unfortunately, we cannot resolve such satellite peaks here for both P1 and P2, likely due to the small CEF splitting and large energy broadening. Therefore, how the modified CEF splitting at the surface affects the P1-P2 transition requires further study in the future.

To clarify this issue, we have made the following changes to the manuscript:

1. (page 5, 3rd paragraph): “While inelastic neutron scattering suggests an excited CEF quartet at ~ 17 meV above the ground state doublet [45], we do not observe any clear satellite in the Kondo resonance peaks, likely due to the large width and limited energy resolution.”
2. (page 6, 3rd paragraph in “Discussion” section): “Theoretically, the reduced layer spacing at the surface reduces the crystal symmetry (from fcc to hcp), which in turn changes the CEF splitting and broadening. The altered CEF states could further affect the Kondo screening and possibly the bandwidth-control delocalization. More studies are needed in the future to resolve the fine structures in the quasiparticle bands associated with the CEF states.”

In Fig.1 P1 phase shows a V like conduction band above -2eV. This band seems to be gone in the P2 phase. The DFT calculations in S2 do not show the disappearance of this band. Can the authors comment on this?

Our response:

We thank the reviewer for this comment. The V-shaped conduction band above -2 eV is indeed not obvious in P2 from Fig. 1 – this is due to the color contrast. Below we replotted Fig. S1 (similar to Fig. 1 but with more data) with enhanced color contrast. Now this V-shaped conduction band can be identified in P2, despite much weaker intensity. The intensity reduction is likely caused by change of the photoemission matrix element as a result of electronic transition. To clarify this issue, we added the following notes in the Fig. S1 caption, which reads: “Note that the V-shaped conduction band at ~ -1.8 eV is still present in P2, although it is barely observable in the current

color contrast.”.

Figure R1. Replot of Fig. S1(a) data with enhanced color contrast. The V-shaped band is highlighted by yellow arrows.

Finally, the emergence of the $f_{5/2}$ and $f_{7/2}$ states in both P1 and P2 phases are a consequence of the Kondo effect, meaning that in both phases there should be some level of localized moments. This may indicate that the system in the P2 phase is not fully itinerant yet.

Our response:

We thank the reviewer for this comment. Indeed, some residual $4f_{5/2}^f$ and $4f_{7/2}^f$ states are present in the P2 phase, implying that the Kondo effect is still in play for this phase. This is consistent with our interpretation of P2 (as discussed earlier): although the bandwidth-control Mott physics plays a dominant role and can explain the the sharp dispersive quasiparticles near E_F , the Kondo effect does coexist and can even cooperate with the Mott physics, leading to complex 4f spectral function and unusual temperature dependence. This is perhaps not surpsing: the coupling between 4f and conduction electrons (essentially the Kondo screening) should always be present, even if (a large part of) the 4f electrons become Mott delocalized due to a larger bandwidth. The coexistence of Kondo screening in P2 provides an interesting opportunity to study the interplay between Mott and Kondo physics.

To further clarify this issue, we have revised the texts accordingly, which includes:

1. (abstract): “which could be a direct consequence of the delicate interplay between the bandwidth-control Mott physics and the coexisting Kondo hybridization.”
2. (“Discussion” section, 1st paragraph): “We mention that some residual Kondo resonance peaks can be observed in P2 (Fig. 4(a) and Fig. 6(b)), indicating that the Kondo effect arising from the Ce^{+3} local moments is still in play here.”

The authors clearly demonstrate that the observed transition is a surface effect and not a property of the bulk. This does not change the importance of the work, however, the authors need to clearly state that in their abstract/title.

Our response:

We thank the reviewer for this suggestion. We have revised the **abstract** to clarify this point, which

now reads: “Here we demonstrate that the bandwidth-control **orbital-selective delocalization** of 4f electrons can be **realized in epitaxial Ce films by thermal annealing, which results in a metastable surface phase with a reduced layer spacing**”. We mention that based on our ex-situ XRD and transport results (see new Fig. S4 and Fig. S6), it is possible to stabilize this metastable surface phase in an appreciable volume, detectable by bulk probes. We have added this point to the revised manuscript, which now reads: “**Finally, the possibility of stabilizing such surface phase in appreciable volume (detectable by bulk-sensitive XRD and transport measurements) offers exciting opportunities in the future to study the physical properties associated with the 4f Mott physics.**” (“discussion” section, end of 3rd paragraph).

The color scale in Fig.1a P0 phase is saturated. Can this be changed? Also, the authors claim that “these states remain localized at each individual Ce site and hence show no momentum Dependence”. However, this is a polycrystalline sample. So what is the size of the ARPES beam and is it expected from a polycrystals to see momentum dependence?

Our response:

We thank the reviewer for the suggestions. We have changed the color scheme in **Fig. 1(a)**, which now has a larger dynamic window and allows for better visualization of the experimental data. We have also changed the statement regarding the P0 phase to avoid possible confusion, which now reads “**The Kondo resonance peaks near E_F do not show any momentum dependence (expected for a polycrystal sample), implying that these resonance states remain localized at each individual Ce site.**” (page 3, 1st paragraph). Here we meant to emphasize that the many-body Kondo resonances can already be observed for a disordered sample, whose spectral function can be well described by the single impurity Anderson model. Experimentally, our ARPES beam spot is $\sim 1 \times 1 \text{ mm}^2$ (**included now in the experimental description**) and it should cover many small domains of the polycrystalline sample (P0). Therefore, the 4f peaks should not have any momentum dependence.

REVIEWER COMMENTS

Reviewer #1 (Remarks to the Author):

I have only one request to the authors in my previous referee report, i.e., to clarify what the authors mean by "localized 4f calculations" and "itinerant 4f calculations". What really distinguished these two types of calculations in practice? According to the authors' reply, these two types of calculations differ simply by the type of pseudopotentials they used. Namely, in the "localized 4f calculations", the 4f electrons are treated as core electrons in the pseudopotential, whereas in the "itinerant 4f calculations", the 4f electrons are explicitly included as the valence electrons.

I am confused. The pseudopotential treatment is merely an approximation, and the ultimate justification of its correctness can only be all-electron calculation. Usually, small-core pseudopotential is more accurate than the large-core pseudopotential because the former is closer to the all-electron results. Ideally, the use of different pseudopotentials shouldn't change the physical results. If this happens, it simply means one or both of the calculations are not reliable, and one should immediately check against all-electron calculations. In the present system, if excluding the 4f electrons from valence electrons yields a qualitatively different physical picture, this indicates one shouldn't do so, period. The different results obtained using different pseudopotentials cannot be used to interpret different physical phases. Just ask what happens if one simply performs an all-electron calculation for the same system?

Reviewer #2 (Remarks to the Author):

I think the authors have fully addressed my previous concerns. So I recommend publication of this work.

Reviewer #3 (Remarks to the Author):

The authors have presented additional data and analysis to tackle all the raised questions. The revised manuscript (and SI) answers all my questions. I therefore recommend the publication of the manuscript in Nature Communications.

Re: NCOMMS-20-38164A

Authors: Yi Wu, *et al.*

Dear reviewers,

Thank you very much for the reports. We appreciate all your comments and suggestions. Below we further address the question raised by reviewer 1 (the revised part in the manuscript is colored in red).

Reviewer 1:

I have only one request to the authors in my previous referee report, i.e., to clarify what the authors mean by "localized 4f calculations" and "itinerant 4f calculations". What really distinguished these two types of calculations in practice? According to the authors' reply, these two types of calculations differ simply by the type of pseudopotentials they used. Namely, in the "localized 4f calculations", the 4f electrons are treated as core electrons in the pseudopotential, whereas in the "itinerant 4f calculations", the 4f electrons are explicitly included as the valence electrons.

I am confused. The pseudopotential treatment is merely an approximation, and the ultimate justification of its correctness can only be all-electron calculation. Usually, small-core pseudopotential is more accurate than the large-core pseudopotential because the former is closer to the all-electron results. Ideally, the use of different pseudopotentials shouldn't change the physical results. If this happens, it simply means one or both of the calculations are not reliable, and one should immediately check against all-electron calculations. In the present system, if excluding the 4f electron from valence electrons yields a qualitatively different physical picture, this indicates one shouldn't do so, period. The different results obtained using different pseudopotentials cannot be used to interpret different physical phases. Just ask what happens if one simply performs an all-electron calculation for the same system?

Our response:

We thank the reviewer for pointing out this confusion. For most simple elements (d-transition metal, s/p-elements), it is true that the choice of core states in pseudopotential does not normally change physical results. But for Ce systems with many-body Kondo physics, the situation is different and the choice of two pseudopotentials (localized and itinerant 4f calculations) actually corresponds to two limits of 4f-electron behaviors. In fact, it is a common practice to perform such calculations for Ce-based systems, see for example PRL 124, 166403 (2020) & PNAS 117, 23467 (2020). The fundamental reason is that DFT cannot correctly deal with localized 4f moments as well as their delocalization in the paramagnetic normal state. Thus, for comparison with the Fermi surface or band structures near the Fermi energy from ARPES, one is forced to consider two limiting situations: one is the itinerant limit, where the 4f electrons are fully delocalized with appreciable bandwidth and Kondo hybridization, and should therefore be treated as valence states; the other is the localized limit, where the 4f electrons are fully (Mott) localized due to strong onsite Coulomb interaction (without Kondo hybridization), cannot be described by the band picture, and thus have to be removed from the band calculations, either by treating them as core electrons (now

the core of Ce pseudopotential is NOT full shell) or simply replacing Ce with La (which inevitably introduces unphysical error). In this way, one may compare with the ARPES data to determine if the Ce 4f electrons are actually localized or itinerant or in between. The two limits are therefore not different solutions of the same setup, but different setups to overcome the limitation of DFT itself.

The issue is similar in VASP (pseudopotential) and WIEN2k (all-electron). For VASP, one chooses different pseudopotentials (treating 4f as valence or core electrons) for Ce in correspondence with the two different physical situations (itinerant versus localized 4f). Both results can be well reproduced from all-electron FLAPW calculations as in WIEN2k, where the 4f electrons are treated either with (for localized case) or without (for itinerant case) “open-core” approximation.

To clarify this point, we have revised the manuscript accordingly, which now reads “**For the itinerant 4f calculations, the Ce-4f electrons are included as the valence states; while for the localized 4f calculations, the Ce-4f states are treated as core electrons, similar to the open-core treatment in full-potential linearized augmented plane-wave (FLAPW) calculations. These two calculations are often carried out in Ce-based systems to represent two limiting situations of Ce-4f electrons (fully itinerant vs localized), see for example [53].**” (Page 7, DFT calculation).